# Octaarginine Improves the Efficacy of Nitazoxanide against *Cryptosporidium parvum*

**DOI:** 10.3390/pathogens11060653

**Published:** 2022-06-06

**Authors:** Tran Nguyen-Ho-Bao, Lum A. Ambe, Maxi Berberich, Carlos Hermosilla, Anja Taubert, Arwid Daugschies, Faustin Kamena

**Affiliations:** 1Centre for Infectious Medicine, Institute of Parasitology, Faculty of Veterinary Medicine, University of Leipzig, 04103 Leipzig, Germany; nhbtran@ctu.edu.vn (T.N.-H.-B.); maxi.berberich@gmx.de (M.B.); daugschies@vetmed.uni-leipzig.de (A.D.); 2Department of Veterinary Medicine, College of Agriculture, Can Tho University, Can Tho 900000, Vietnam; 3Laboratory for Molecular Parasitology, Department of Microbiology and Parasitology, University of Buea, Buea P.O. Box 63, Cameroon; lumabienwiambe@yahoo.com; 4Centre for Research on Health and Priority Pathologies, Institute of Medical Research and Medicinal Plants Studies (IMPM), Yaoundé P.O Box 13033, Cameroon; 5Biomedical Research Center Seltersberg (BFS), Institute of Parasitology, Justus Liebig University Giessen, 35392 Giessen, Germany; carlos.r.hermosilla@vetmed.uni-giessen.de (C.H.); anja.taubert@vetmed.uni-giessen.de (A.T.)

**Keywords:** *Cryptosporidium*, nitazoxanide, octaarginine

## Abstract

Cryptosporidiosis is an intestinal disease that affects a variety of hosts including animals and humans. Since no vaccines exist against the disease till date, drug treatment is the mainstay of disease control. Nitazoxanide (NTZ) is the only FDA-approved drug for the treatment of human cryptosporidiosis. However, its efficacy in immunocompromised people such as those with AIDS, in malnourished children, or those with concomitant cryptosporidiosis is limited. In the absence of effective drugs against cryptosporidiosis, improving the efficacy of existing drugs may offer an attractive alternative. In the present work, we have assessed the potential of the cell-penetrating peptide (CPP) octaarginine (R8) to increase the uptake of NTZ. Octaarginine (R8) was synthetically attached to NTZ in an enzymatically releasable manner and used to inhibit growth of *Cryptosporidium parvum* in an in vitro culture system using human ileocecal adenocarcinoma (HCT-8) cell line. We observed a significant concentration-dependent increase in drug efficacy. We conclude that coupling of octaarginine to NTZ is beneficial for drug activity and it represents an attractive strategy to widen the repertoire of anti-cryptosporidial therapeutics. Further investigations such as in vivo studies with the conjugate drug will help to further characterize this strategy for the treatment of cryptosporidiosis.

## 1. Introduction

*Cryptosporidium* spp. is a small protozoan parasite that infects both animals and humans worldwide, causing gastroenteritis. Owing to the wide range of hosts and various transmission pathways, *Cryptosporidium parvum* has not only a tremendous impact on animal health but also needs to be considered within the context of the “One Health” concept. Cryptosporidiosis in humans with an intact immune system is self-limiting and is clinically manifested by transient diarrhea, nausea, vomiting, fever, and abdominal pain [1,2]. In contrast, the infection of immunocompromised patients or malnourished children with either *C. parvum* or *Cryptosporidium hominis* may become life threatening without proper treatments [3,4]. Though the introduction of highly active antiretroviral therapy (HAART) against HIV infection has significantly reduced the incidence of fatal cryptosporidiosis in industrialized countries [5,6], *Cryptosporidium* infection was still listed in 2013 by the Global Enteric Multicenter Study (GEMS) as the second leading cause of diarrhea-associated mortality in children in developing countries [4]. *C. parvum* is zoonotic and can cause severe disease in both humans and animals, particularly so in young ruminants. So far, there are no vaccines available against cryptosporidiosis and therapeutic options to treat and control the disease are rather scanty. Nitazoxanide (NTZ) is the only FDA-approved drug for chemotherapeutic treatment of human cryptosporidiosis. However, NTZ shows poor efficacy in immunocompromised patients and malnourished children [7,8] and is not licensed for use in animals. Thus, the development of new or alternative therapeutic strategies to control cryptosporidiosis is mandatory. However, developing new drugs is a time-consuming and costly process. Alternatively, repurposing drugs that are established for other indications or increasing the efficacy of available anti-cryptosporidial compounds represents attractive options for widening the repertoire of therapeutic solutions. 

In recent years, the short polycationic peptide octaarginine (R8) has been successfully used as a delivery vehicle for drugs, as well as plasmid DNA, into intracellular parasites [9,10,11]. It has been demonstrated that octaarginine significantly improves the efficacy of the experimental anti-malarial drug fosmidomycin [10]. In this study, we investigated the potential of this short cell-penetrating peptide octaarginine as a delivery tool to increase uptake of NTZ into *C. parvum*-infected cells, and the resulting increase in concentration and efficacy. For this purpose, octaarginine was coupled to NTZ to create the NTZ-R8 conjugate. Importantly, the octaarginine moiety was coupled to NTZ via an ester bond that can be released by intracellular esterases, and the conjugated compound was tested in vitro. 

Here, we show that NTZ linked to a carrier octaargenine was delivered more to intracellular compartments harboring *C. parvum* stages (i. e. trophozoites and meronts) and was more efficacious in inhibiting *C. parvum* growth in in vitro human ileocecal adenocarcinoma (HCT-8) cell line. Taken together, coupling of NTZ to a carrier octaarigine molecule resulted in increased drug uptake and inhibitory efficacy in vitro.

## 2. Results

### 2.1. Uptake of FAM-Labelled Octaarginine by Cryptosporidium parvum

6-FAM-labeled octaarginine was supplemented to either freshly excysted *C. parvum* sporozoites or *C. parvum*- infected HCT-8 cell cultures. For extracellular sporozoites, the peptide uptake occurred rather rapidly, with octaarginine accumulating in the parasite nucleus within 10 min (Figure 1a). The incubation of FAM-labelled octaarginine with *C. parvum*-infected HCT-8 monolayers revealed that octaarginine entered the parasitophorous vacuole (PV) of all intracellular stages (i.e., sporozoites) within 1 h of incubation (Figure 1b). These observations confirm the potential of octaarginine to pass across membranes into both extracellular and intracellular stages of *C. parvum*.

### 2.2. NTZ-R8 Inhibits Intracellular Cryptosporidium parvum Growth in HCT-8 Cells

Nitazoxanide was coupled synthetically to octaarginine to generate the conjugate nitazoxanide-octaarginine (NTZ-R8). An ester bond was used for the coupling of the two moieties, allowing for the hydrolysis and release of the active tizoxanide moeity by esterases (Figure 2).

Before evaluating the activity of NTZ-R8 on the target organism, *C. parvum*, potential cytotoxic effects of NTZ and octaarginine on HCT-8 host cells were evaluated using the MTT test. Concentration ranges of 1 to 25 µg/mL for NTZ and 1 to 100 µg/mL for octaarginine were tested. No relevant cytotoxicity was found for both substances. However, cell viability was slightly reduced by 14.99% and 6.6% at the very high and non-physiological doses of NTZ and CPP, respectively (Figure 3).

To test for the inhibitory properties of the drugs, HCT-8 cells were infected with *C. parvum* sporozoites and treated with NTZ or NTZ-R8 at six different concentrations (1, 5, 10, 50, 100, 1000 ng/mL). These treatment doses have been previously tested by MTT assay and shown to have no toxicity on host cells. At the highest concentration of 1000 ng/mL, NTZ-R8 and NTZ distinctly inhibited intracellular *C. parvum* replication by 97.57% and 79.98%, respectively (Figure 4).The IC_50_ was 60.54 ng/mL (197 nM) for NTZ and 4.499 ng/mL (2.9 nM) for NTZ-R8 (Figure 4). The coupling of NTZ to octaarginine (NTZ-R8) thus significantly increased the inhibitory effect of NTZ on *C. parvum* growth in vitro (*p* = 0.0045, paired *t*-test).

## 3. Discussion

Nitazoxanide (NTZ) is the only FDA-approved drug for the treatment of cryptosporidiosis in humans. However, NTZ is not efficacious in cryptosporidiosis treatment of malnourished children and HIV patients. In particular, NTZ showed limited efficacy in cryptosporidiosis treatment of HIV patients as demonstrated in different studies [8,12,13] and in immunocompromised mice [14]. To improve its efficacy, NTZ was coupled to the short polycationic cell-penetrating peptide octaarginine. NTZ is a prodrug that is rapidly hydrolyzed by esterase and transformed into its diacetyl derivative, tizoxanide, which is the active metabolite [15]. Considering this feature, octaarginine was coupled to NTZ by an ester bond and designed to release tizoxanide upon esterase cleavage (Figure 3) in the cell. Octaarginine has been shown to cross various biological membranes such as the blood-brain barrier [16] and membranes of different cell types [9,17]. Therefore, to evaluate the potential use of octaarginine as a delivery vehicle for NTZ in cryptosporidiosis treatment, we first assessed the permeation of octaarginine across parasite and cell membranes. The limiting factor for cell-penetrating peptides such as octaarginine to function as a carrier for any cargo into a cell is the permeability across the corresponding biological membrane, and the process is independent of the cargo size [9,18]. FAM-labelled octaarginine was used to track the permeation of this small peptide through the parasite as well as host membranes. We found that octaarginine was able to penetrate all parasite stages, although the free sporozoites accumulated the peptides much faster than the intracellular meronts. This is probably the result of the multiple membrane barriers existing around the intracellular parasite at its extracytoplasmic location [19]. Interestingly, although *Cryptosporidium* species are obligate intracellular parasites, they never enter host cytosol, but remain throughout their entire intracellular life in an extracytoplasmic space directly underneath the host plasma membrane [20]. Our findings show that octaarginine penetrates all membranes surrounding the parasite with a faster kinetic than it penetrates the host cell membrane. These results suggested that coupling of NTZ to octaarginine represents an attractive and novel alternative to increase NTZ accumulation in the parasite. 

Concordant with the above-mentioned results on the permeability of parasite membrane to octaarginine, we observed during the in vitro experiments that NTZ coupled with octaarginine (NTZ-R8) inhibits the growth of *C. parvum* in HCT-8 cells more efficiently than NTZ alone after 24 h of exposure. The calculated IC_50_ of NTZ- R8 was 2.9 nM while that of NTZ alone was 197 nM. By facilitating the transport of NTZ across the host cell membrane and subsequently through the parasite membrane, octaarginine improved the activity of NTZ by more than 60-fold. In our study, NTZ at a dose of 1000 ng/mL inhibits *C. parvum* replication by 79%, a rate that is higher than previously published data [20,21,22]. Although NTZ can be dissolved in DMSO, we recognized that precipitation occurred after further dilution in DMEM culture medium. Therefore, we modified the protocol by ultrasonication with amplitude 60% for 30 s (two to three times) (Bandelin sonopuls GM70, Berlin, Germany) to ensure complete dissolution in the medium before adding to the infected host cells. This modification of the protocol routinely used in other laboratories might have led to a slight improvement of drug solubility.

Although many drugs have been shown to successfully inhibit *Cryptosporidium* spp. in vitro, e.g., monensin, halofuginone, paromomycin [21,23], the inhibition in animal models was less obvious [13]. Indeed, many variables such as bioavailability, pharmacokinetics, food-drug interaction [24] influence the efficacy in animals and could partly explain the difficult transferability of in vitro results to in vivo conditions [25]. 

Altogether, we can conclude that coupling of NTZ to octaarginine significantly improves the drug efficacy in vitro. The improvement observed is very likely the result of more efficient uptake of the passenger drug into infected host cells due to efficient carriage by octaarginine. This is in line with previous observations in other intracellular parasitic disease models (e.g., *Plasmodium* spp.) [10] and emphasizes the potential use of cell-penetrating peptides to act as a delivery vehicle for anti-parasitic drugs. 

## 4. Materials and Methods

### 4.1. Compounds

Nitazoxanide (NTZ) (Sigma-Aldrich, Darmstadt, Germany) and nitazoxanide-octaarginine (NTZ-R8) were dissolved in DMSO (10 mg/mL; Merck, Darmstadt, Germany) and stored in the dark at −20 °C until use. NTZ-R8 was custom-synthesized by JPT peptide (Berlin, Germany). The drugs were freshly prepared in infection medium [DMEM supplemented with 2% fetal calf serum, 1% antibiotics penicillin/streptomycin, and 1% amphotericin B (all Sigma-Aldrich, Darmstadt, Germany)] for in vitro testing.

### 4.2. Synthesis of Octaarginine-6-FAM

Octaarginine-6-FAM was synthesized as previously published [10].

### 4.3. Cell Culture

Permanent human ileocecal adenocarcinoma cells (HCT-8) were seeded into 24-well plates at 2 × 10^5^ cells/well. The cells were grown in RPMI medium (Sigma-Aldrich, Darmstadt, Germany) to 70–80% confluence in 1–2 days. The growth medium consisted of RPMI medium supplemented with 10% fetal calf serum (FCS) (Northumbria, Cramlington, UK), 1% antibiotics (penicillin/streptomycin) and 1% amphotericin B (Sigma-Aldrich, Darmstadt, Germany).

### 4.4. Parasites

The *Cryptosporidium parvum* subtype used was IIaA15G2R1 (AB560747), which is the hypertransmissible one and the most frequent in both ruminants and humans. *C. parvum* oocysts were passaged every 3 months in calves under experimental conditions and oocysts were purified from feces following the protocol described by [26]. Oocysts were stored at 4 °C in PBS, pH = 7.2 (Gibco, ThermoFisher Scientific, Waltham, MA, USA) supplemented with penicillin/streptomycin (200 µg/mL) and amphotericin B (5 µg/mL) to prevent bacterial and fungal growth for up to 3 months until use. The storage medium was replaced every 2 weeks. Before usage, the oocysts were bleached with cold NaOCl (5.25% diluted 1:1 (*v*/*v*) in PBS; pH = 7.2) by incubating on ice for 5 min. Oocysts were then washed extensively with cold PBS (3 times) to completely remove NaOCl before sporozoite excystation. To obtain free-released sporozoites, oocysts were resuspended in excystation medium (sodium taurocholate-NaT at a final concentration of 0.4% in DMEM medium supplemented with 2% FCS, 1% penicillin/streptomycin, 1% amphotericin B, and 1% of sodium pyruvate) and processed following a standard protocol as described in [11].

### 4.5. Uptake of FAM-Labeled Octaarginine by Excysted Sporozoites and Intracellular Cryptosporidium parvum

6-FAM-labeled octaarginine (10 μg/mL) was added to either free sporozoites or HCT-8 cell cultures infected by intracellular stages of *C. parvum*. Intracellular 6-FAM-labeled octaarginine was visualized by direct fluorescence microscopy and immunofluorescence assay was applied to detect *C. parvum* using a Leica TCS SP8 laser scanning confocal microscope (Leica, Wetzlar, Germany). Sporozoites were centrifuged at 9500× *g* for 5 min, followed by a washing step with PBS (pH = 7.2). All steps of the following protocol were performed at room temperature (RT). Sporozoites or infected host cells were fixed with 4% paraformaldehyde (PFA) for 20 min, and thereafter washed 3 times with PBS. Then, 4,6-diamidino-2-phenylindole (DAPI; 10 µg/mL) was added followed by incubation for another 5 min. In the immunofluorescence assay, permeabilization with 0.2% Triton X-100 for 20 min was performed right after the fixation step. Then, 1% bovine serum albumin (BSA) in PBS was added to block unspecific binding. Thereafter, infected host cells were incubated for 1 h with a specific primary rat-anti *Cryptosporidium* antibody (Waterborne INC, New Orleans, LA, USA) in PBS containing 1% BSA at a dilution of 1: 1000. Goat-anti-rat Dylight 647 (Rockland, Limerick, ME, USA) was used as a secondary antibody and the nuclei were stained with DAPI. Finally, cells were mounted with Fluoromount-G (Southern Biotech, Birmingham, AL, USA) and stored at 4 °C until visualization.

### 4.6. Mitochondrial Toxicity Test (MTT)

The MTT 3-(4,5-dimethythiazol2-yl)-2,5-diphenyl tetrazolium bromide (Sigma-Aldrich, Darmstadt, Germany) assay was used to evaluate cell viability in media with NTZ and octaarginine at different concentrations. A total of 7 × 10^4^ HCT-8 cells were seeded into 96-well plates and incubated at 37 °C under 5% CO_2_ until the cell cultures reached 80% confluency. NTZ (25, 20, 15, 10, 5, 1 µg/mL) and octaarginine (100, 10 and 1 µg/mL) were freshly diluted in culture medium and added to the growing cultures for 24 h. Subsequently, 10 µL MTT solution (containing tetrazolium dye) was added to each well and the plates were further incubated for 4 h. Stop solution (10% SDS/0.01 M HCl) was added to dissolve precipitates of formazan crystals. Absorption was measured by spectrophotometry at 595 nm. Each experiment was carried out in triplicates. 

### 4.7. In Vitro Inhibition Assay

HCT-8 cells were seeded into 24-well plates (2 × 10^5^ cells/well) and incubated until they reached 80% confluency. Cells were cultured in RPMI-1640 medium supplemented with 10% FCS, antibiotics (1% penicillin/streptomycin, 1% amphotericin B), and 1% sodium pyruvate. They were incubated at 37 °C with 5% CO_2_. Confluent monolayers were inoculated with 2 × 10^5^ freshly excysted sporozoites in infection medium (DMEM with 2% FCS, 1% amphotericin B and 1% penicillin/streptomycin, 1% sodium pyruvate) and further incubated at 37 °C and 5% CO_2_ for 3 h. Non-excysted oocysts and empty oocyst shells were gently removed by washing with PBS 3 times. NTZ and NTZ-R8 were diluted in growth medium at different concentrations (1, 5, 10, 50, 100, 1000 ng/mL), added to infected cultures and further incubated (37 °C, 5% CO_2_) for 24 h. Uninfected untreated cells served as negative controls. Positive controls were *C. parvum* infected cultures that were not treated with any drug. All experiments were carried out in triplicates. 

### 4.8. RNA Extraction

Exactly 24 h post-infection, cell culture plates were centrifuged at 1000*× g* for 10 min to ensure that the remaining extracellular stages of the parasite firmly settle on the bottom of the well. The culture medium was gently aspirated and cells were harvested by directly adding lysis buffer of the RNeasy Mini Kit (Qiagen, Hilden, Germany) to each well. Further extraction of RNA from the samples was carried out strictly following the instructions of the manufacturer. Total RNA was measured by a NanoPhotometer NP80 (Implen, Munich, Germany). 1 µg RNA was used to produce cDNA according to the instruction delivered with the Revert-Aid first-strand cDNA synthesis kit (Thermo Fisher Scientific, Darmstadt, Germany). The cDNA was stored at −80 °C until use.

### 4.9. Real-Time PCR

Real-time PCR reactions were performed on a Bio-Rad CFX96 Touch Real-Time PCR Detection^®^ system using the program two-step SYBR green with primers for *Cryptosporidium* 18S RNA targeting Cp18S-1011F (5′-TTG TTC CTT ACT CCT TCA GCA C-3′) and Cp18S-1185R (5′- TCC TTC CTA TGT CTG GAC CTG-3′). The data were normalized by the transcription levels of host cell Hs18S rRNA [27]., applying the primer pair Hs18S- 1F (5′-GGC GCC CCC TCG ATG CTC TTA-3′) and Hs18S- 1R (5′-CCC CCG GCC GTC CCT CTT A-3′). The thermocycler program for RT-PCR was: 95 °C for 3 min, followed by 40 amplification cycles at 95 °C for 10 s and 58 °C for 30 s. Melting curve analysis was performed at a temperature range between 65 °C and 95 °C. The transcription level of Hs18 rRNA was applied for both normalization and controls. The following formulae were used to estimate parasite growth inhibition (PGI%)
(1)$$\Delta C_{T\;}=C_{T\left[\mathrm{Cp}18S\right]\;}-C_{T\left[\mathrm{Hs}18S\right]\;}$$
(2)$$\Delta \Delta C_{T\;}=\Delta C_{T\left[\mathrm{sample}\right]\;}-\Delta C_{T\left[\mathrm{control}\right]}$$
(3)$$\mathrm{PGI}\%=100-\left(2^{-△△C_{T}}\right)\times 100$$


### 4.10. Data Analysis

Data obtained by qPCR were analyzed using Microsoft Excel^®^ (Microsoft Corporation, Redmond, WA, USA). Statistical analyses and graphs were carried out using GraphPad Prism^®^ version 8.02 (GraphPad Software, Inc., La Jolla, CA, USA). IC_50_ values were generated following a logarithmic plot of data from the dose-response inhibition experiment. D’Agostino-Pearson normality test was used to ascertain the normal distribution of data. Differences were considered statistically significant when *p* < 0.05

## Figures and Tables

**Figure 1 pathogens-11-00653-f001:**
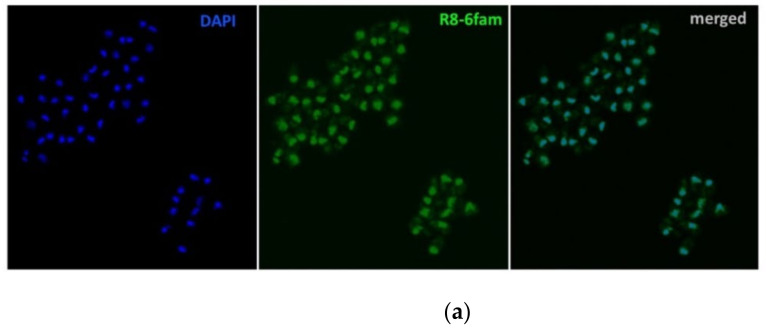
Uptake of FAM-labelled octaarginine by *Cryptosporidium parvum*. (**a**) Freshly excysted *C. parvum* sporozoites were incubated with 6FAM -octaarginine (R8-6FAM) for 10 min before analysis and visualization under a fluorescence microscope. (**b**) Logarithmic in vitro culture of *C. parvum* in HCT-8 cells were incubated with 6FAM-octaarginine for 1 h before analysis. Octaarginine accumulates in the host cell nucleus but also in the intracellular parasite (arrows). Sporo-Glo (red) was used for the specific visualization of *C. parvum* and DAPI was used to visualize the DNA.

**Figure 2 pathogens-11-00653-f002:**
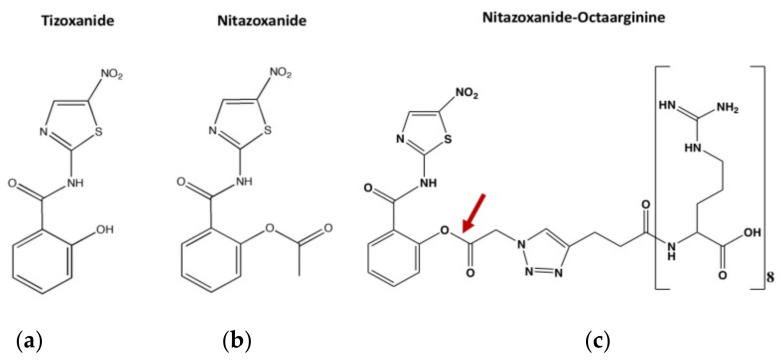
Chemical structures of tizoxanide, nitazoxanide and nitazoxanide-octaarginine. (**a**) Tizoxanide is a metabolite of Nitazoxanide, (**b**) Nitazoxanide, (**c**) Nitazozanide-octaarginine was synthesized by coupling the cell-penetrating peptide octaarginine with nitazoxanide. The two moieties were linked via a cleavable ester bond (red arrow) to enable release of active tizoxanide moiety following cleavage by esterases.

**Figure 3 pathogens-11-00653-f003:**
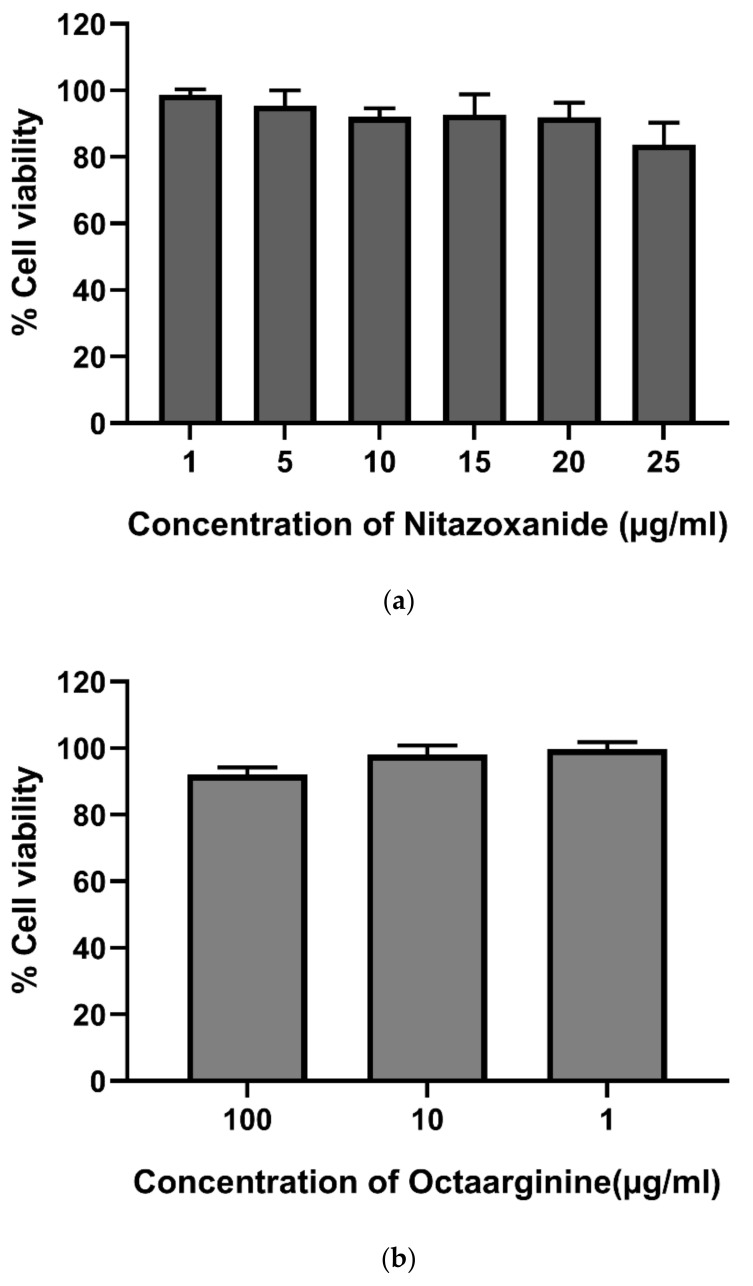
Viability of HCT-8 cells following exposure to NTZ (**a**) and octaarginine (**b**) throughout 24 h of drug exposure.

**Figure 4 pathogens-11-00653-f004:**
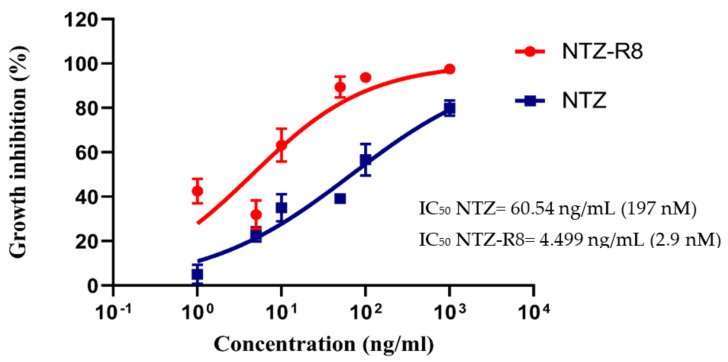
Effect of NTZ and NTZ-R8 on intracellular *Cryptosporidium parvum*. Growth inhibition of *C. parvum* in HCT-8 monolayers by NTZ and NTZ-R8. Mean values with standard errors from three replicates were plotted. The value of IC_50_ was calculated by Graphpad^®^ 8.1: *p* < 0.01, paired *t*-test.

## Data Availability

Not applicable.

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
