# Peer review of "Octaarginine Improves the Efficacy of Nitazoxanide against Cryptosporidium parvum"

_pathogens, 2022, doi:10.3390/pathogens11060653_

Round 1

Reviewer 1 Report

Dears, Good day,

Original research topic. I have a few revisions:

Line 23: replace « new » by « effective »

Line 25: “octaarginine (R8)  to increase the uptake of NTZ.  Octaarginine (R8)  was ….”

Line 165: sonication (which power?), which apparatus?

Line 198: The most important revision: please justify the use  of the IIaA15G2R1 C. parvum subtype. Please add that this is the hypertransmissible one, and the most frequent  in both ruminants and humans.

Best regards,

Reviewer 2 Report

Dear authors 

 The manuscript “Octaarginine improves the efficacy of nitazoxanide against

Cryptosporidium parvum” is well written manuscript. Authors did a great work on improving the efficacy of Nitazoxanide which is the only FDA- approved drug for Cryptosporidiosis. Authors investigated the ability of the cell-penetrating peptide octaarginine to enhance the uptake of NTZ. Authors showed that coupling of octaarginine to NTZ is important for drug activity and thus serve as a better anti-cryptosporidial therapy.  There remain some minor issues that authors should consider the comments useful for further revision of the manuscript.

  Comments:

Line 105 -106: However, cell viability was slightly reduced at the very high and non-physiological doses of both chemicals (Figure 3). Authors need to show what percentage reduced?

Line 254: Authors indicate that Positive controls were C. parvum infected cultures that were not treated with any drug. Authors need to use another known drug control which kills the parasite  

Figure 1. Needs a good resolution figure, especially Fig 1.b

Figure 3. There are some problems with legends which are not readable

Authors suggests that coupling of NTZ to octaarginine significantly improves the drug efficacy in vitro.

Figure 4. Authors show the IC50 of NTZ and NTZ-R8. What is the selectivity index of these drugs ? . It would be interesting to find out the selectivity index

Reviewer 3 Report

Comments to the Author

The manuscript entitled "Octaarginine Improves the Efficacy of Nitazoxanide against Cryptosporidium parvum" represents a considerable amount of work. The following comments need to be addressed before the manuscript is suitable for publication in Pathogens Journal.

  • In material and methods, if you have an accession number for this isolate please, added it.
  • What is the concentration of oocysts used?
